# The Association Between Fast Food Consumption and Inflammatory Bowel Disease: A Case-Control Study and Meta-Analysis

**DOI:** 10.3390/nu17111838

**Published:** 2025-05-28

**Authors:** Anas Almofarreh, Haytham A. Sheerah, Ahmed Arafa, Abdulatif M. AlBassam, Mshari A. Alassaf, Faisal M. AlBassam, Faisal B. Alsaif, Khalid M. Alkwai, Faisal A. Alzahrani, Mohammed A. Allift, Shahad AlBassam, Aseel AlBassam, Mohammed Alshehri, Khalid O. Alshammari, Njoud M. Alenezi, Fahad A. Alamri

**Affiliations:** 1Assistant Deputyship for Health Investment Development, Ministry of Health, Riyadh 11451, Saudi Arabia; 2The Advisory Unit, Ministry of Health, Riyadh 11451, Saudi Arabia; 3Department of Preventive Cardiology, National Cerebral and Cardiovascular Center, Suita 564-8565, Japan; ahmed011172@med.bsu.edu.eg; 4Department of Public Health and Community Medicine, Faculty of Medicine, Beni-Suef University, Beni-Suef 62521, Egypt; 5Preventive Medicine, Armed Forces Hospital Southern Region, Khamis Mushait 62413, Saudi Arabia; abalbasssam@gmail.com; 6Department of Family Medicine, Security Forces Hospital, Riyadh 12625, Saudi Arabia; misharyaalassaf@gmail.com; 7Department of Family Medicine, King Fahad Medical City, Riyadh 11525, Saudi Arabia; faisal-515@hotmail.com (F.M.A.);; 8Department of Anatomical Pathology, King Khalid University Hospital, Riyadh 12372, Saudi Arabia; 9Faculty of Medicine, Dar Al Uloom University, Riyadh 13314, Saudi Arabia; 10Assistant Deputyship for Medical Assistance Services, Ministry of Health, Riyadh 11451, Saudi Arabia; 11Family Medicine Department, Hail Health Cluster, Hail 55471, Saudi Arabia; 12Global Centre for Mass Gatherings Medicine, Ministry of Health, Riyadh 12372, Saudi Arabia

**Keywords:** Crohn’s disease, fast food, ulcerative colitis, meta-analysis

## Abstract

**Background:** Inflammatory bowel diseases (IBD), including ulcerative colitis (UC) and Crohn’s disease (CD), are chronic conditions influenced by various factors, including diet. This study examined the association between fast food consumption and IBD risk through a case-control study and a meta-analysis of epidemiological evidence. **Methods:** We analyzed data from a hospital-based case-control study conducted in Riyadh. The study included 158 UC patients, 244 CD patients, and 395 controls without IBD. Fast food consumption was assessed using a self-administered questionnaire distributed before diagnoses were made. We used logistic regression to calculate odds ratios (ORs) and 95% confidence intervals (95% CIs) of UC and CD for individuals who reported daily fast food consumption. Then, we merged our results with those from other studies investigating the same association into a meta-analysis. **Results:** In the case-control study, daily consumption of fast food was strongly associated with UC and CD among Saudi people: age- and sex-adjusted ORs (95% CIs) = 6.29 (3.89, 10.16) and 5.92 (3.98, 8.80), respectively. The associations remained robust after further adjustments: ORs (95% CIs) = 6.61 (3.93, 11.12) and 5.90 (3.89, 8.94), respectively. Similarly, the meta-analysis revealed higher odds of fast food intake associated with UC and CD, with pooled odds ratios (95% CIs) of 2.41 (1.07, 5.45) and 2.65 (1.23, 5.70), respectively. **Conclusions:** Our findings highlight the potential role of fast food consumption in the development of IBD. From a preventive medicine perspective, fast food consumption should be discouraged to reduce the risk of IBD.

## 1. Introduction

Inflammatory bowel disease (IBD), which includes ulcerative colitis (UC) and Crohn’s disease (CD), is a chronic inflammatory disorder that affects the gastrointestinal tract. The disease is associated with debilitating symptoms, including abdominal pain, diarrhea, weight loss, and fatigue [1]. IBD has a profound psychosocial impact, often leading to anxiety, depression, and substantial reductions in patients’ quality of life [2,3]. Epidemiological evidence indicates that the burden of IBD has risen worldwide, particularly in newly industrialized countries undergoing rapid lifestyle and dietary changes. From 1990 to 2019, 13 out of the 21 regions defined by the Global Burden of Disease (GBD) study experienced a rise in the age-standardized prevalence rate of IBD. By 2019, the global numbers of IBD patients and IBD-related deaths were 4,900,000 and 41,000, respectively, and the age-standardized prevalence, death, and disability-adjusted life-years (DALYs) rates were 59.25 (52.78, 66.47), 0.54 (0.46, 0.59), and 20.15 (16.86, 23.71), respectively [4]. Additionally, IBD has a hefty financial burden. A review of 638,664 IBD patients in the United States estimated direct annual costs per patient, including outpatient, inpatient, and pharmacy expenses, between $7824 and $41,829 [5].

The precise cause of IBD is still not fully understood, but it is thought to arise from a complex interplay of genetic predisposition, environmental influences, and immune system dysregulation [1]. However, dietary factors have been increasingly implicated in the pathogenesis of IBD [6]. Among these factors, fast food consumption, characterized by high levels of saturated fats, refined carbohydrates, emulsifiers, and preservatives, is associated with gut dysbiosis, increased intestinal permeability, and systemic inflammation, all of which can contribute to IBD development [7,8]. Experimental studies suggest that food additives and high-fat diets may disrupt gut homeostasis and enhance intestinal inflammation [9,10]. Still, epidemiological evidence on the association between fast food and IBD is limited and inconsistent [11].

Conversely, research exploring the link between dietary factors and IBD is limited in the Arab populations. Since most existing studies originate from Western countries, it remains unclear whether their conclusions apply to regions with distinct sociocultural contexts, such as Saudi Arabia. IBD has been increasingly reported in Saudi Arabia over the previous decades [12,13]. This rise coincides with a significant shift in dietary habits driven by urbanization, economic growth, and greater exposure to global food markets. Traditional diets rich in whole grains, legumes, and fresh products have been increasingly replaced by fast food, mirroring dietary trends observed in Western nations [14,15].

Given the increasing prevalence of IBD and the shifting dietary habits in Saudi Arabia, we analyzed data from a case-control study conducted in the country to explore the association between fast food consumption and IBD. We also systematically reviewed existing epidemiological evidence on this association and combined the results of the case-control study with those from previous studies in a meta-analysis.

## 2. Methods

### 2.1. The Case-Control Study

#### 2.1.1. Study Design and Population

The study population comprised individuals aged 18 years and older who were diagnosed at a private polyclinic in Riyadh, Saudi Arabia, between January 2009 and December 2017. Using a convenient sampling approach, this case-control study included 171 patients with UC, 251 with CD, and 400 individuals with other gastrointestinal conditions who served as the control group. Eligibility for the UC and CD groups required a recent diagnosis, while controls were selected based on the absence of IBD, malignancy, polyposis, and diverticulosis. Participants lacking data on fast food consumption were excluded, resulting in a final analytical sample of 158 UC patients, 244 CD patients, and 395 controls.

#### 2.1.2. Exposure, Outcome, and Covariates

Fast food consumption was collected via a self-administered questionnaire completed before the IBD diagnosis was made. The question used to assess fast food intake was: “How often do you eat fast food?” with response options including “once or less/month”, “once/week”, “twice/week”, “every day”, “do not remember”, and “not applicable”. Participants who selected “do not remember” or “not applicable” were excluded. Participants were instructed to report their usual dietary habits based on a typical month, excluding periods of fasting such as Ramadan. Illiterate participants were assisted by trained data collectors.

The diagnosis of IBD was established based on a standardized diagnostic protocol, consistent with previously published studies [16,17,18,19]. All patients presenting with gastrointestinal symptoms underwent a thorough clinical evaluation, which included detailed medical history, physical examination, and laboratory investigations. These investigations comprised blood tests and both urine and stool analyses, aimed at identifying markers of inflammation and excluding infectious or other non-inflammatory causes of gastrointestinal symptoms. Patients with clinical and laboratory findings suggestive of IBD underwent further diagnostic assessment using lower gastrointestinal endoscopy. High-definition video endoscopes from Olympus, Pentax, or Fujinon were employed to conduct a thorough inspection of the intestinal mucosa. During the procedure, targeted mucosal biopsies were obtained and subsequently evaluated by experienced pathologists through histopathological examination to confirm the presence of IBD and differentiate between UC and CD.

Information on age, sex, and smoking behavior was obtained from the same questionnaire. Data on anemia (Hemoglobin < 13.0 g/dL in men and < 12.0 g/dL in women) and elevated liver enzyme levels (ALT and/or AST > 50 U/L) were extracted from blood test results. Body mass index (BMI) was measured at the clinic after questionnaire collection. BMI categories were as follows: Underweight < 18.5 kg/m^2^, normal weight 18.5–24.9 kg/m^2^, overweight 25.0–29.9 kg/m^2^, and obesity ≥ 30.0 kg/m^2^. All participants declined alcohol consumption; therefore, this variable was not included in the analysis.

#### 2.1.3. Study Size

Details of the sample size determination have been published in prior studies [16,17,18,19].

#### 2.1.4. Statistical Analysis

We used logistic regression analysis to calculate odds ratios (ORs) and their corresponding 95% confidence intervals (CIs) for the association between fast food consumption and the risk of UC and CD. Adjustments were made for age, sex, BMI, smoking status, anemia, and liver enzyme levels. Interaction analyses were conducted to examine the effect of age and sex on the association between fast food consumption and the risk of UC and CD. Statistical analyses were conducted using IBM SPSS Statistics for Windows, Version 22.0 (IBM Corporation, Armonk, NY, USA).

### 2.2. The Meta-Analysis

#### 2.2.1. Registration and Protocol

This meta-analysis followed the Preferred Reporting Items for Systematic Reviews and Meta-Analyses (PRISMA) guidelines [20]. The study protocol was registered with the PROSPERO International Prospective Register of Systematic Reviews (ID: 1006388).

#### 2.2.2. Eligibility Criteria

Studies were eligible for inclusion in this meta-analysis if they met the following criteria: (1) fast food consumption was the exposure; (2) UC or CD was the outcome; (3) the study reported risk estimates or prevalence data related to UC or CD across categories of fast food consumption; and (4) the study was published in English. We excluded duplicates, irrelevant articles, animal studies, uncontrolled studies, editorials, abstracts without full texts, review articles, case reports, and any articles that did not meet our inclusion criteria.

#### 2.2.3. Information Sources

We searched the Medline (PubMed), Web of Science, and Scopus databases for relevant literature. The search was limited to studies published before 1 March 2025. We also manually screened the reference lists of included articles and relevant review papers to identify additional eligible studies.

#### 2.2.4. Search Strategy

Two authors independently conducted comprehensive research using predefined search terms relevant to fast food consumption and IBD: ((Fast food) OR (Unhealthy Diet) OR (Junk food)) AND ((Inflammatory bowel disease) OR (Ulcerative Colitis) OR (Crohn’s disease)).

#### 2.2.5. Selection Process

Titles and abstracts retrieved from the initial search were independently screened by two authors. Full texts of potentially eligible studies were then reviewed and assessed for inclusion based on predefined criteria. Any disagreements were resolved through discussion and, when necessary, consultation with other authors.

#### 2.2.6. Data Collection Process

Two authors independently extracted data from the included studies using a standardized form. Discrepancies were addressed through consensus and, if needed, in consultation with a third author.

#### 2.2.7. Data Items

Extracted data included the publication year, geographic location, sample size, study design, measures of fast food consumption, controlled covariates, and risk estimates related to UC or CD.

#### 2.2.8. Study Risk of Bias Assessment

The quality of included studies was evaluated using a modified version of the Newcastle–Ottawa Scale (NOS) [21]. The criteria assessed included case and control definitions, representativeness of samples, comparability of study groups, assessment of fast food consumption, consistency of data collection methods across groups, and response rates. Two authors conducted the assessments independently, and disagreements were resolved through discussion.

#### 2.2.9. Effect Measures

We extracted and pooled odds ratios (ORs) with corresponding 95% confidence intervals (CIs) to quantify the association between fast food consumption and the risk of UC or CD.

#### 2.2.10. Synthesis Methods

We used a random-effects model to compute pooled ORs and 95% CIs, comparing the highest vs. lowest levels of fast food consumption [22]. Heterogeneity was assessed using τ^2^ (total heterogeneity), I^2^ (proportion of total variability due to heterogeneity), and H^2^ (ratio of total variability to sampling variability) statistics [23].

#### 2.2.11. Reporting Bias Assessment

We evaluated the possibility of publication bias by conducting a regression test for funnel plot asymmetry [24]. All statistical analyses were conducted using the R 3.2.0 statistical software package (Metafor: Meta-Analysis Package for R) [25].

#### 2.2.12. Certainty Assessment

We assessed the certainty of evidence using the Grading of Recommendations Assessment, Development, and Evaluation (GRADE) approach across five domains: risk of bias, inconsistency, indirectness, imprecision, and publication bias.

## 3. Results

### 3.1. The Case-Control Study

#### 3.1.1. Participants and Descriptive Data

Compared to the control group, UC patients had lower proportions of obesity (31.4% vs. 22.2%; *p*-value < 0.05) and current smoking (20.8% vs. 11.4%; *p*-value < 0.05) but a higher prevalence of anemia (15.4% vs. 43.7%; *p*-value < 0.05). Similarly, CD patients had lower obesity rates (31.4% vs. 9.8%; *p*-value < 0.05) and a higher prevalence of anemia (15.4% vs. 22.1%; *p*-value < 0.05). Sex distribution did not differ significantly between UC and CD patients and their respective controls (*p*-value > 0.05). However, CD patients were significantly younger than both their controls and UC patients (*p*-value < 0.05) (Table 1).

#### 3.1.2. Associations

In the regression model adjusted for age and sex, daily consumption of fast food was associated with UC and CD: ORs (95% CIs) = 6.29 (3.89, 10.16) and 5.92 (3.98, 8.80), respectively. Further adjustments for BMI, smoking, anemia, and liver enzymes did not significantly change the results: ORs (95% CIs) = 6.61 (3.93, 11.12) and 5.90 (3.89, 8.94), respectively (Table 2). Sex and age did not impact the relationship between fast food consumption and IBD (Interaction *p*-values > 0.10).

### 3.2. The Meta-Analysis

#### 3.2.1. Study Selection

After removing duplicates, reviews, and studies with unrelated exposures or outcomes, four studies were identified for inclusion in the meta-analysis (Figure 1). These were combined with the findings from our case-control study, resulting in a total of five eligible studies.

#### 3.2.2. Study Characteristics

Among the five studies included, four employed case-control designs, while one used a cross-sectional design. The included studies were published between 1992 and 2021 and investigated the relationship between fast food consumption and either ulcerative colitis (UC) or Crohn’s disease (CD) (Table 3).

#### 3.2.3. Quality Assessment

Using the modified Newcastle–Ottawa Scale (NOS), all included studies were rated as having moderate quality (Appendix A). Risk of bias primarily stemmed from the assessment of dietary exposure and comparability between cases and controls.

#### 3.2.4. Results of Individual Studies

For the analysis of UC, the contribution of each study to the overall weight was as follows: Qualqili et al. (19.8%) [26], DeClercq et al. (20.7%) [27], Niewiadomski et al. (20.6%) [28], Persson et al. (17.3%) [29], and Almofarreh et al. (current case-control study) (21.6%). Niewiadomski et al. [28], Persson et al. [29], and Almofarreh et al. (current case-control study) studies revealed a positive association between fast food consumption and UC, while Qualqili et al. [26] and DeClercq et al. [27] did not reach the same conclusion. In the analysis of CD, four studies were included, with the following weights: DeClercq et al. (24.4%) [27], Niewiadomski et al. (27.5%) [28], Persson et al. (20.2%) [29], and Almofarreh et al. (current case-control study) (27.9%). Only DeClercq et al. [27] did not find a positive association between fast food consumption and CD.

#### 3.2.5. Results of Syntheses

The pooled analysis showed a statistically significant association between fast food consumption and UC, with an OR of 2.41 (95% CI: 1.07, 5.45) (Figure 2). Substantial heterogeneity was detected across studies (τ^2^ = 0.729, I^2^ = 86.06%, H^2^ = 7.17). Sensitivity analyses, conducted by excluding one study at a time, did not significantly reduce heterogeneity (Appendix A). Similarly, the meta-analysis of studies on CD revealed a significant association with fast food consumption, yielding an OR of 2.65 (95% CI: 1.23, 5.70) (Figure 3). High heterogeneity was also observed (τ^2^ = 0.504, I^2^ = 85.73%, H^2^ = 7.01), and sensitivity analyses did not substantially influence these results (Appendix A).

#### 3.2.6. Publication Bias

No evidence of publication bias was found in either analysis. For UC, the regression test for funnel plot asymmetry yielded z = –0.272 (*p* = 0.786) (Appendix A). For CD, the test result was z = –0.393 (*p* = 0.694) (Appendix A).

#### 3.2.7. Certainty of Evidence

The GRADE assessment indicated an overall low certainty of evidence (Appendix A). This rating was based on the observational nature of the included studies. The certainty was further limited by a potential bias due to self-reported dietary data and incomplete adjustment for confounders in some studies, as well as inconsistency stemming from variation in fast food definitions and categorization and recall periods. However, no serious concerns were identified for indirectness, as the populations and exposures were relevant, and imprecision was not serious, given that all studies reported odds ratios and confidence intervals. Additionally, publication bias was unlikely, as the funnel plot showed no significant asymmetry.

## 4. Discussion

This study provides compelling evidence of a positive association between fast food consumption and the risk of UC and CD. Our case-control study conducted in Saudi Arabia demonstrated that individuals with higher fast food intake had an increased likelihood of developing IBD. These findings were further reinforced by our meta-analysis, which extended the association to both Western and Arab populations, suggesting that the impact of fast food on IBD risk is consistent across diverse dietary and cultural settings.

Several biological mechanisms may explain the observed relationship between fast food consumption and IBD. Fast foods are typically rich in trans fats, refined sugars, and food additives, which can promote intestinal inflammation, alter gut microbiota composition, and compromise the gut barrier. Additionally, a diet high in ultra-processed foods (UPFs) and low in fiber may reduce short-chain fatty acid production, which plays a protective role in gut health [7,8,9,10]. Previous meta-analyses suggested that dietary habits and specific foods linked to fast food consumption are associated with an elevated risk of IBD. A meta-analysis of observational studies indicated that a Western dietary pattern was associated with an increased risk of UC (relative risk (RR): 2.05; 95% CI: 1.32, 3.18) and CD (RR: 1.65; 95% CI: 1.02, 2.85) [11]. Another meta-analysis of cohort studies reported a higher risk of CD development among individuals with greater UPF consumption compared to those with lower intake (hazard ratio: 1.68; 95% CI: 1.34, 2.10) [30]. A dose–response meta-analysis indicated that for every 10% increase in daily UPFs intake, the risk of CD rose by 18% (RR: 1.18; 95% CI: 1.02, 1.30) [31]. Two meta-analyses identified significant associations between soft drink consumption and an increased risk of UC (RR: 1.65; 95% CI: 1.22, 2.25) [32] and CD (RR: 1.39; 95% CI: 1.03, 1.95) [33]. Another meta-analysis showed a significant protective effect of fruit intake against UC (RR: 0.69; 95% CI: 0.55, 0.86) and CD (RR: 0.47; 95% CI: 0.38, 0.58). Likewise, vegetable consumption was significantly inversely associated with the risk of UC (RR: 0.56; 95% CI: 0.48, 0.66) and CD (RR: 0.52; 95% CI: 0.46, 0.59) [34].

Apart from IBD, frequent fast food consumption has been linked to several health complications, including obesity, cardiovascular diseases, diabetes, metabolic syndrome, hypertension, dyslipidemia, and non-alcoholic fatty liver disease [35]. Additionally, it has been associated with an increased risk of depression, stress, cognitive decline, and poor mental health outcomes [36].

From a public health perspective, our findings highlight the urgent need for dietary interventions with the aim of reducing fast food consumption, particularly among populations undergoing rapid nutritional transitions. Public awareness campaigns, policy initiatives, and healthcare strategies promoting healthier dietary patterns may help mitigate the growing burden of IBD.

## 5. Strengths and Limitations

This case-control study possesses several notable strengths. First, it addresses a significant knowledge gap by focusing on an understudied population in the Arab region, where evidence on the association between dietary habits and IBD is limited. Second, the study employed standardized and clinically recognized diagnostic criteria for both UC and CD, ensuring diagnostic accuracy and consistency. Third, including newly diagnosed patients likely reduces the risk of recall bias, as participants were asked about their dietary habits within a relatively short period before diagnosis, minimizing the possibility of inaccurate or retrospective reporting. These strengths enhance the relevance and internal validity of the study’s findings. However, several limitations should also be acknowledged. First, data were collected from a single private clinic in Riyadh, which may limit the generalizability of the results to other regions or populations with different healthcare access or dietary practices. Second, the study’s recruitment spanned an extended time frame, during which fast food availability and consumption patterns may have changed, potentially affecting the accuracy of exposure classification. Third, the food frequency questions used to assess fast food intake were not validated or pre-tested. Fourth, fast food consumption was measured as a general category without distinguishing between specific types of fast food or accounting for nutritional components such as saturated fat, sodium, or caloric content. Fifth, due to the small number of cases in each frequency category, participants with non-daily intake were grouped, which prevented a detailed analysis of a possible dose–response relationship. Sixth, the study did not account for several potential confounding variables that could have influenced the observed associations. These include levels of physical activity [37], family history of IBD, presence of comorbid conditions such as diabetes and metabolic syndrome [38], and the use of medications, particularly antibiotics and corticosteroids, which are known to affect gut health and may alter the risk of IBD [39]. Moreover, the analysis did not adjust for total caloric intake [6] and other dietary factors, especially those characterizing the market of fast food in Saudi Arabia, which is often linked to lower overall dietary quality. This dietary pattern is generally characterized by a higher consumption of carbohydrates and free sugars, along with a reduced intake of fiber, dairy products, fruits, and vegetables [40]. Previous studies conducted among Saudi patients have reported a positive association between the consumption of sugary beverages and the risk of IBD, while higher intake of fruits, vegetables, and dairy products was found to be negatively associated with IBD [16,17,18,19]. Seventh, we do not have enough data about IBD activity; therefore, we were not able to investigate the association between fast food consumption and IBD activity.

The primary strength of the meta-analysis was the ability to increase the number of UC and CD cases by incorporating data from multiple studies and investigating populations representing different regions. However, several limitations should be acknowledged. First, the meta-analysis included a limited number of studies; therefore, we could not stratify the meta-analysis by region, study design, or frequency of fast food consumption. Second, the high heterogeneity across studies is another limitation. It could be attributed to several factors, including differences in study design, sample sizes, dietary assessment methods, and population characteristics. Variability in how fast food consumption was categorized across studies may have led to inconsistencies in exposure classification. Additionally, the included studies covered different geographical regions, with variations in dietary patterns, lifestyle factors, and genetic predispositions that could influence IBD risk. Differences in adjustment for covariates may have also contributed to heterogeneity. Third, key confounding factors, including total energy intake and lifestyle variables, were not adjusted for across studies.

## 6. Conclusions

Our findings indicate a significant association between fast food consumption and an increased risk of UC and CD. This association was observed both in the case-control study conducted among Saudi individuals and in the meta-analysis, which included populations from Western and Arab regions. Given the rising prevalence of fast food consumption worldwide, these findings highlight the need for public health initiatives aimed at promoting healthier dietary habits to reduce the burden of IBD. Future research, particularly prospective cohort studies, is required to establish a causal link and further explore the underlying mechanisms driving this association. Studying the association between dietary factors and IBD activity and severity should be considered as well.

## Figures and Tables

**Figure 1 nutrients-17-01838-f001:**
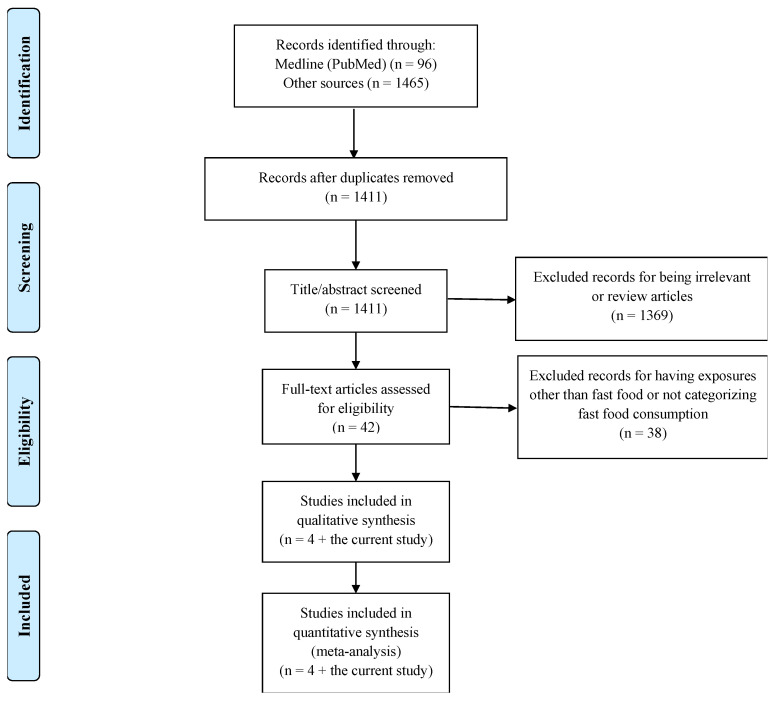
PRISMA flowchart of the studies included in the meta-analysis.

**Figure 2 nutrients-17-01838-f002:**
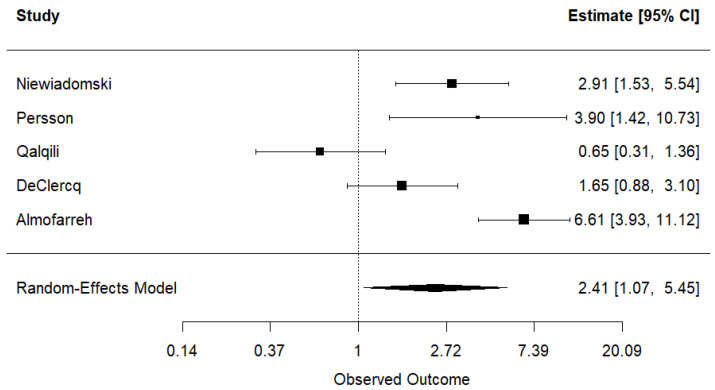
Meta-analysis of the association between fast food consumption and ulcerative colitis [26,27,28,29].

**Figure 3 nutrients-17-01838-f003:**
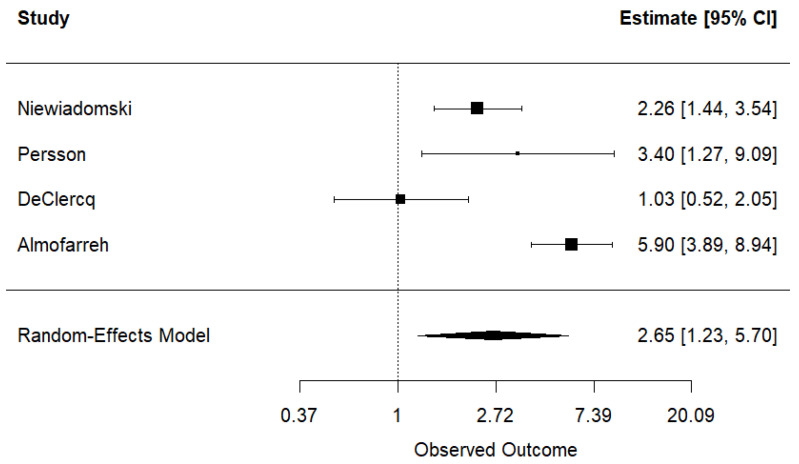
Meta-analysis of the association between fast food consumption and Crohn’s disease [27,28,29].

**Table 1 nutrients-17-01838-t001:** Comparison between cases and controls.

Characteristics		Ulcerative Colitis	Crohn’s Disease	Control
Number of Participants		158	244	395
Age, years, %	<30	20.3	33.2	17.7
	30–39	34.8	44.3	43.0
	≥40	44.9	22.5	35.3
Sex, %	Men	58.9	68.0	64.8
	Women	41.1	32.0	35.2
Body mass index, kg/m^2^, %	<18.5	8.9	18.4	9.4
	18.5–24.9	45.6	51.2	30.4
	25.0–29.9	23.4	20.5	28.9
	≥30	22.1	9.9	31.3
Current smoking, %		11.4	18.9	20.8
Anemia, %		43.7	22.1	15.4
Elevated liver enzymes, %		15.2	11.1	19.0

**Table 2 nutrients-17-01838-t002:** Association between fast food consumption and inflammatory bowel disease in the case-control study.

Fast Food Consumption	Ulcerative Colitis *n* = 158	Control *n* = 395	Model I	Model II
Daily, %	50.5%	19.7%	6.29 (3.89, 10.16)	6.61 (3.93, 11.12)
Infrequent, %	49.5%	80.3%	1 (Reference)	1 (Reference)
Fast food consumption	Crohn’s disease	Control	Model I	Model II
Daily, %	61.3%	19.7%	5.92 (3.98, 8.80)	5.90 (3.89, 8.94)
Infrequent, %	38.7%	80.3%	1 (Reference)	1 (Reference)

Model I: ORs (95% CIs) adjusted for age and sex. Model II: ORs (95% CIs) adjusted for age, sex, BMI, smoking, anemia, and liver enzymes.

**Table 3 nutrients-17-01838-t003:** Summary of the studies included in the meta-analysis.

Study ID	Study Design	Population	Fast Food Assessment Method, Categories, and Retrospective Period	Covariates or Matched Variables
Almofarreh (2025) Saudi Arabia	Case-control	158 UC patients, 244 CD patients, and 395 without IBD (≥18 years) from a private clinic in Riyadh	Self-administered questionnaire Fast food (daily vs. infrequent) A few weeks before diagnosis	Age, sex, BMI, smoking, anemia, and liver enzymes
Qualqili (2021) Jordan [26]	Case-control	100 UC patients, 85 CD patients, and 150 without IBD (18–68 years) from the University of Jordan Hospital, Zarqa Governmental Hospital, and Al Bashir Hospital	Interview Fast food (1–3 times per week vs. infrequent) 3 months before diagnosis	Age and marital status
DeClercq (2018) Canada [27]	Cross-sectional	119 UC patients, 111 CD patients, and 12,462 without IBD (30–74 years) from the Atlantic Partnership for Tomorrow’s Health study	Self-administered questionnaire Fast food (2–5 times/week vs. no) 1 year before enrolment	Age, sex, and residence
Niewiadomski (2016) Australia [28]	Case-control	51 UC patients, 81 CD patients, and 104 without IBD (11–76 years) from specialists, hospitals, pharmacies, and pathology centres in Melbourne, Victoria	Self-administered questionnaire Fast food (yes vs. no) 6 months before diagnosis	None
Persson (1992) Sweden [29]	Case-control	181 UC patients, 184 CD patients, and 390 without IBD (15–79 years) from hospital admissions of Stockholm County	Self-administered questionnaire Fast food (≥2 times/week vs. no) 1–4 years before diagnosis	Age and sex

## Data Availability

We did not make our data publicly available due to ethical or administrative reasons. However, the corresponding author may provide the original dataset upon a reasonable request after consulting the KFMC IRB.

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
