# Peer review of "The Association Between Fast Food Consumption and Inflammatory Bowel Disease: A Case-Control Study and Meta-Analysis"

_nutrients, 2025, doi:10.3390/nu17111838_

Round 1

Reviewer 1 Report

Comments and Suggestions for Authors

The study “The association between fast food consumption and inflammatory bowel disease: a case–control study and meta–analysis” by Almofarreh et al. addresses an important question in an under-studied Arab population, and the combination of a primary case-control investigation with a meta-analytic synthesis is both interesting and valuable. However, some points need to be addressed.

 The manuscript could benefit from clearer formatting  in the Methods section. In addition, on page 19 the meta-analysis mentions that GRADE assessment was not carried out; if such an evaluation was considered and deemed unnecessary, the authors should state their rationale, or else include a brief GRADE summary to allow readers to judge confidence in the findings.

Furthermore, several potential confounding factors deserve explicit attention in the text. Although the population is predominantly Arab, alcohol use, where present, should be documented and discussed; similarly, participants should have been asked about a family history of inflammatory bowel diseases, including ulcerative colitis and Crohn’s disease, and this variable should be accounted for in the analysis. Medication use, particularly antibiotics that may alter the gut microbiota, ought to have been recorded and considered as a possible modifier of IBD risk. The presence of comorbid conditions such as celiac disease, metabolic syndrome or diabetes mellitus in the study population could also influence results and should be reported or adjusted for. Given regional dietary traditions, it would be helpful to comment on whether fast food in Arab countries, both in terms of ingredients (for example use of lamb, tahini or local spices) and cooking methods, differs from Western-style fast food and how those differences might affect inflammation. Finally, intermittent fasting practices such as Ramadan observance could substantially alter eating patterns and inflammatory status; the authors should clarify whether participants were assessed for fasting, especially during data collection or whether analyses were timed to avoid the fasting month.

Author Response

The study “The association between fast food consumption and inflammatory bowel disease: a case–control study and meta–analysis” by Almofarreh et al. addresses an important question in an under-studied Arab population, and the combination of a primary case-control investigation with a meta-analytic synthesis is both interesting and valuable. However, some points need to be addressed.

Response: Thank you for your valuable feedback. We modified the manuscript accordingly.

The manuscript could benefit from clearer formatting in the Methods section.

Response: We formatted the Methods section as suggested and added many details. The methods have 2 main sections: one section for the case-control study (study design and population, setting, description of exposure, outcome, and covariates assessment, study size, and statistical analysis) and the other section for the meta-analysis (registration and protocol, eligibility criteria, information sources, search strategy, selection process, data collection process, data items, study risk of bias assessment, effect measures, synthesis methods, reporting of bias assessment, and certainty assessment).

In addition, on page 19 the meta-analysis mentions that GRADE assessment was not carried out; if such an evaluation was considered and deemed unnecessary, the authors should state their rationale, or else include a brief GRADE summary to allow readers to judge confidence in the findings.

Response: We agree with you. Therefore, we modified the manuscript as follows: 1) a certainty assessment section was added to the Methods section (2.2.12.), a certainty of evidence section in the Results section (3.3.6.), and a supplementary Table (2) summarizing the GRADE assessment.

Furthermore, several potential confounding factors deserve explicit attention in the text. Although the population is predominantly Arab, alcohol use, where present, should be documented and discussed; similarly, participants should have been asked about a family history of inflammatory bowel diseases, including ulcerative colitis and Crohn’s disease, and this variable should be accounted for in the analysis. Medication use, particularly antibiotics that may alter the gut microbiota, ought to have been recorded and considered as a possible modifier of IBD risk. The presence of comorbid conditions such as celiac disease, metabolic syndrome or diabetes mellitus in the study population could also influence results and should be reported or adjusted for.

Response: We acknowledge these limitations. 1) Alcohol: all participants declined alcohol consumption; therefore, this variable was not involved in the analysis. We added this clarification to the Methods section (2.1.3.). Family history, medication use, diabetes, and metabolic syndrome: We acknowledged the lack of data about these factors in the limitation section.

Given regional dietary traditions, it would be helpful to comment on whether fast food in Arab countries, both in terms of ingredients (for example use of lamb, tahini or local spices) and cooking methods, differs from Western-style fast food and how those differences might affect inflammation.

Response: Previous Saudi studies have reported that the fast food market in the country is quite similar to that in Western countries. This dietary pattern is generally characterized by a higher consumption of carbohydrates and free sugars, along with a reduced intake of fiber, dairy products, fruits, and vegetables. We added these clarifications to the limitation section.

Finally, intermittent fasting practices such as Ramadan observance could substantially alter eating patterns and inflammatory status; the authors should clarify whether participants were assessed for fasting, especially during data collection or whether analyses were timed to avoid the fasting month.

Response: Participants were instructed to report their usual dietary habits based on a typical month, excluding periods of fasting such as Ramadan. We clarified this point in the Methods section (2.1.3.).

Reviewer 2 Report

Comments and Suggestions for Authors

First, I would like to point out that the title of this paper is not formatted correctly, and the term "running title" is not addressed in the guidelines.

Regarding the abstract, I understand the author's confusion about the "multi-paragraph" format. However, the MDPI style guide specifically requires a single-paragraph abstract, as confirmed by sample articles from previous issues. This confusion may have partially arisen due to the unspecified article type. While the introduction section is appropriate and adequately outlines the problem, the methodology section is extremely flawed and lacking.

The authors repeatedly describe this study as "a case-control study" supplemented by a meta-analysis. However, I have concerns about this characterization. While the meta-analysis segment is clear and addresses numerous unanswered questions, the connection between these components is not evident, and the rationale behind the paper remains ambiguous.

Furthermore, the methodology of the "case-control study" lacks crucial information; specifically, the questionnaire is not mentioned until section "2.1.4. Data sources and measurements" and is not adequately described at that point either. Could you please provide the total number of participants, including the numbers of UC and CD patients?

In section 2.1.4, the paragraph beginning with "Information on age, sex, and smoking behavior was obtained from the same questionnaire..." should be separated into its own section.

For a case-controlled study with a meta-analysis, a strict yet comprehensive methodology, an effective evidence collection process, and a detailed description of the meta-analysis process are required. Moreover, this study appears to overlook the risk of bias entirely, which could enhance the value of this paper.

In this study, the limitations are not adequately acknowledged. It is important to provide examples and evidence to support claims about the effects of these limitations, avoiding excuses or exaggerations of their impact. Overall, maintain transparency and objectivity when presenting the limitations, ensuring that the significance of the research remains unchanged intact.

Comments on the Quality of English Language

 I can barely follow the authors' flow of thought.

Author Response

First, I would like to point out that the title of this paper is not formatted correctly, and the term "running title" is not addressed in the guidelines.

Response: We revised the manuscript title. Many Journals ask for a running title during submission to help editors and reviewers identify and manage manuscripts and organize files, even if it doesn’t appear in the final publication.

Regarding the abstract, I understand the author's confusion about the "multi-paragraph" format. However, the MDPI style guide specifically requires a single-paragraph abstract, as confirmed by sample articles from previous issues. This confusion may have partially arisen due to the unspecified article type.

Response: The multi-paragraph format of our abstract is similar to that published in MDPI journals. We conducted a case-control study followed by a meta-analysis. These are a few examples of similar study designs (PMID: 37150604; PMID: 39203883; and PMID: 34666857), and we have published a study with a similar design in Nutrients.

While the introduction section is appropriate and adequately outlines the problem, the methodology section is extremely flawed and lacking. The authors repeatedly describe this study as "a case-control study" supplemented by a meta-analysis. However, I have concerns about this characterization. While the meta-analysis segment is clear and addresses numerous unanswered questions, the connection between these components is not evident, and the rationale behind the paper remains ambiguous.

Response: The rationale behind the paper was explained over two sections in the introduction section, which the reviewer just described as adequately outlining the problem. Still, we may agree with the need for more clarification about the methods section, which we provided in the revised version.

Furthermore, the methodology of the "case-control study" lacks crucial information; specifically, the questionnaire is not mentioned until section "2.1.4. Data sources and measurements" and is not adequately described at that point either. Could you please provide the total number of participants, including the numbers of UC and CD patients?

Response: As described in the methods section, we added references to previous studies where the questionnaire was described. The questionnaire itself was provided in previous studies as a supplement. However, we added more details about it in the Methods section. The total number of participants was described as well.

In section 2.1.4, the paragraph beginning with "Information on age, sex, and smoking behavior was obtained from the same questionnaire..." should be separated into its own section.

Response: We modified the entire methods section.

For a case-controlled study with a meta-analysis, a strict yet comprehensive methodology, an effective evidence collection process, and a detailed description of the meta-analysis process are required. Moreover, this study appears to overlook the risk of bias entirely, which could enhance the value of this paper.

Response: We modified the entire methods section.

In this study, the limitations are not adequately acknowledged. It is important to provide examples and evidence to support claims about the effects of these limitations, avoiding excuses or exaggerations of their impact. Overall, maintain transparency and objectivity when presenting the limitations, ensuring that the significance of the research remains unchanged intact.

Response: We added more explanations to the limitations section

Reviewer 3 Report

Comments and Suggestions for Authors

Nutrients-3649383
Title: The association between fast food consumption and inflammatory bowel disease: a case-control study and meta-analysis, by Anas Almofarreh: 

GENERAL COMMENTS:
The authors analyzed data on IBD cases including various bowel diseases to know the impact of fast food consumption.  They found that, in the case-control study, daily consumption of fast food was strongly associated with UC and CD among Saudi people.  The meta-analysis also revealed higher odds of fast food intake associated with UC and CD.  The authors concluded that, from a preventive medicine perspective, discouraging fast food consumption should be considered to reduce the risk of IBD.  The points of study are interesting and well-focused.  However, the several issues are raised to improve this study.

SPECIFIC COMMENTS:
1)    The diagnosis of IBDs and the pathological confirmation should be clarified in the present subjects.  The duration of IBDs and the prescription and the duration of steroids and anti-inflammatory agents should also be addressed to know the impact of food taking on the IBD conditions.
2)    How were the usage of glucocorticoids in these diseases?  How about the age when diagnosed as IBDs ?  The duration and the family history of IBDs are also important to know the patients’ lifestyles.
3)    Laboratory biomarkers indicating the activities of each IBD would be included to clarify the conditions.  Some representative case presentations showing the effects of food taking may also be informative in this study.
4)    As the authors mentioned in the limitation, key confounding factors, including total energy intake and lifestyle variables, should be carefully adjusted between the related studies.
5)    The discussion should be included the related key references including the heterogeneity of the study population of meta-analyses.  Also, the more profound discussion on the mechanism by which the fast food and IBD activity are associated should be considered.
6)    Addressing the quality of these data and the impact of the data on the findings of the reviews and meta-analyses will improve the quality of the interpretations in this manuscript by utilizing the established quality assessment tools for the different research designs by the EQUATOR websites.

Author Response

GENERAL COMMENTS:
The authors analyzed data on IBD cases including various bowel diseases to know the impact of fast food consumption.  They found that, in the case-control study, daily consumption of fast food was strongly associated with UC and CD among Saudi people.  The meta-analysis also revealed higher odds of fast food intake associated with UC and CD.  The authors concluded that, from a preventive medicine perspective, discouraging fast food consumption should be considered to reduce the risk of IBD.  The points of study are interesting and well-focused.  However, the several issues are raised to improve this study.

Response: Thank you for your valuable comments. We modified the manuscript accordingly.

SPECIFIC COMMENTS:
1)    The diagnosis of IBDs and the pathological confirmation should be clarified in the present subjects.  The duration of IBDs and the prescription and the duration of steroids and anti-inflammatory agents should also be addressed to know the impact of food taking on the IBD conditions.

Response: IBD diagnosis: We added detailed explanations for the diagnosis of IBD in the Methods section (2.1.3.). Duration of IBD and medications: All patients were newly diagnosed. The questionnaire was distributed and collected before the IBD diagnosis was made. We clarified both points in the Methods section (2.1.1 and 2.1.2).
2)    How were the usage of glucocorticoids in these diseases?  How about the age when diagnosed as IBDs ?  The duration and the family history of IBDs are also important to know the patients’ lifestyles.

Response: All patients were newly diagnosed. The questionnaire was distributed and collected before the IBD diagnosis was made. We clarified both points in the Methods section (2.1.1 and 2.1.2). Regarding the lack of data about IBD history and medications, we acknowledged this point in the limitations section of the discussion.
3)    Laboratory biomarkers indicating the activities of each IBD would be included to clarify the conditions.  Some representative case presentations showing the effects of food taking may also be informative in this study.

Response: We did not collect enough data about IBD activity and grading. Additionally, this study aimed to describe the association between fast food consumption and IBD rather than IBD activity. However, we agree with the Reviewer that examining the association with IBD activity and severity would be a great idea. We added it to the future directions in the conclusion section.
4)    As the authors mentioned in the limitation, key confounding factors, including total energy intake and lifestyle variables, should be carefully adjusted between the related studies.

Response: We agree with you. We added a detailed description of the potential confounders to the limitation section.
5)    The discussion should be included the related key references including the heterogeneity of the study population of meta-analyses.  Also, the more profound discussion on the mechanism by which the fast food and IBD activity are associated should be considered.

Response: We expanded the description of the heterogeneity across studies in the limitations section. We also expanded the discussion on the potential mechanisms in the discussion section.
6)    Addressing the quality of these data and the impact of the data on the findings of the reviews and meta-analyses will improve the quality of the interpretations in this manuscript by utilizing the established quality assessment tools for the different research designs by the EQUATOR websites.

Response: We modified the manuscript as follows: 1) a certainty assessment section was added to the Methods section (2.2.12.), a certainty of evidence section in the Results section (3.3.6.), and a supplementary Table (2) summarizing the GRADE assessment.

Round 2

Reviewer 1 Report

Comments and Suggestions for Authors

The authors have sufficiently addressed the points raised in the previous review round.

Reviewer 2 Report

Comments and Suggestions for Authors

This version, like the previous one, lacks line numbers, which makes peer review difficult. Additionally, the definition of patients serving as controls needs to be clarified. For example, why were polyposis and diverticulosis considered exclusion criteria? A diagram illustrating the patient selection process should also be included. I'm curious—how many participants were temporarily excluded from the dietary monitoring?

Author Response

This version, like the previous one, lacks line numbers, which makes peer review difficult.

Response: We added line numbers.

Additionally, the definition of patients serving as controls needs to be clarified. For example, why were polyposis and diverticulosis considered exclusion criteria?

Response: We clarified the definition of the control group. Excluding individuals with malignancy, polyposis, and diverticulosis from the control group was essential to minimize confounding and enhance the internal validity of the study. These conditions, like IBD, involve pathological changes in the GIT and may influence dietary patterns, particularly the intake of fast food, which was a key exposure variable in the study. Including such individuals could obscure the association between diet and IBD by introducing overlapping symptoms or altered nutritional behaviors unrelated to IBD itself. Moreover, diverticulosis and colorectal polyps may be associated with chronic inflammation or structural changes in the colon, which could mimic or confound IBD pathology. Therefore, their exclusion ensures a more appropriate comparison group.

A diagram illustrating the patient selection process should also be included. I'm curious—how many participants were temporarily excluded from the dietary monitoring?

Response: The study population comprised individuals aged 18 years and older who were diagnosed at a private polyclinic in Riyadh, Saudi Arabia, between January 2009 and December 2017. Using a convenient sampling approach, this case-control study included 171 patients with UC, 251 with CD, and 400 individuals with other gastrointestinal conditions who served as the control group. Eligibility for the UC and CD groups required a recent diagnosis, while controls were selected based on the absence of IBD, malignancy, polyposis, and diverticulosis. Participants lacking data on fast food consumption were excluded, resulting in a final analytical sample of 158 UC patients, 244 CD patients, and 395 controls (lines 88-95).

Reviewer 3 Report

Comments and Suggestions for Authors

The authors have appropriately revised their manuscript according to the referees' comments.

Author Response

The authors have appropriately revised their manuscript according to the referees' comments.

Response: Thank you.